# Training Freshmen Engineers as Managers to Develop Soft Skills: A Person-Centred Approach

Rosa-María Rodríguez-Jiménez [1], Pedro J. Lara-Bercial [1] and María-José Terrón-López [2,*]

1 Science, Computation and Technology Department, School of Architecture, Engineering and Design, Universidad Europea de Madrid, 28670 Villaviciosa de Odón, Spain; rosamaria.rodriguez@universidadeuropea.es (R.-M.R.-J.); pedro.lara@universidadeuropea.es (P.J.L.-B.)
2 Aerospace and Industrial Engineering Department, School of Architecture, Engineering and Design, Universidad Europea de Madrid, 28670 Villaviciosa de Odón, Spain
* Correspondence: m_jose.terron@universidadeuropea.es

**Abstract:** This article describes a subject design to train engineer students in soft skills through an experiential and person-centred approach, as is usually developed in companies for managers to incorporate responsible and ethical engineering perspectives. This design is based on an experiential methodology and its impact on students is presented. The subject, called "Personal and Professional Effectiveness", was designed and developed based on Covey's classic book "The 7 habits of highly effective people" and introduces ethical and social commitment among other contents. A detailed description of different activities carried out within the module is provided to allow replication. Students' perception, gathered from their reflexive diaries, and academic satisfaction survey results are presented. Students report that this teaching method leads to a deeper connection with themselves and increased awareness of their strengths and weaknesses. Students seem to have understood the importance of sustainability and how individual behavior impacts in an engineering team. They reflect on how being aware of their own strengths and difficulties helps them integrate different knowledge into their daily lives and how this can improve their behaviors, not only professionally, but also personally.

**Keywords:** experiential learning; freshmen; 7 habits; soft skills; personal & professional development

## 1. Introduction

In 2004, the 'Shanghai Declaration on Engineering and the Sustainable Future' [1] called for promoting engineering for a sustainable future. Engineers must not only acquire technical knowledge but a range of soft skills to achieve better performance in their professional career [2–4]. In fact, engineers' technical work cannot be separated from the soft skills to achieve sustainability as approved by the Millennium Development Goals [5] followed by the UN Sustainable Development Goals (SDGs) for 2030. In particular the SDG4, "Quality Education" plays an important role in accomplishing the 2030 agenda. We must train engineering students oriented towards employability, but reminding them that they must be responsible, ethical, and aware of the contribution that engineering can make to societies and the quality of life [4,6–8]. The European Higher Education Area approach encourages student-centred learning [9] and the inclusion of soft skills in the curricula [10,11], paying attention to the ones formulated by UNESCO [8]. The inclusion of sustainability into engineering education is a reality in many universities even though there is still room to design the learning activities that enable students to achieve the related learning outcomes [12,13]. Therefore, it has been a shift in engineering education during the past 100 years changing the focus of how the content should be taught [14]. In an attempt to follow these guidelines, our engineering school decided to move towards a project-based learning methodology in all its degrees [15]. The essence of Project Based Learning is the setting of an initial question that serves to organize and

drive the learning activities and culminates in a student-generated product or output that answers the question. Teaching staff play an advisory role encouraging students to be proactive. So, developing key skills and competences in students is needed to effectively design and develop the projects, as engineers' technical work is inseparably of generic competences [6]. This need to guarantee that every student gained the required soft skills was driven by including, as Martin [16] suggests, three transversal subjects throughout the curriculum of each degree (1st year: Personal and Professional Effectiveness; 2nd year: Impact and Influence of Relationships and 3rd year: Entrepreneurial Leadership). So, while this article is geared toward engineering education, the ideas here could be applied to any degree.

Reflective, experiential and person-centred practice is fundamental in developing soft skills [17–19]. Moreover, in most companies, managers are trained experientially using a person-centred approach [20]. The person-centred concept stems from Carl Rogers' humanistic psychology. Rogers [21] highlights the importance of working with the individual's subjective experience wherein self-awareness is considered as the basis for developing someone's full potential. Personal factors that are part of self-awareness such as attitudes, values, and beliefs seem to be the best predictors of assuming a sustainable behavioral intention, as pointed out by different studies [22,23]. Although intention alone does not always lead to consistent behavior because it is influenced by the context or situation [24] it is the place from which to start working towards sustainable behaviors [25]. On the other hand, the implementation of learning strategies based on multiple intelligences [26] improves the interest and motivation of students by improving their self-esteem [27].

In parallel, creating an emotionally positive atmosphere in the classroom encourages motivation and learning [28]. Experiential activities give rise to dynamic emotions such as surprise and joy, and consequently produce a bigger commitment to the subject [29].

The paper would be guided by the following question: is it possible to train engineer students in soft skills using an experiential and person-centred approach as it is usually developed in companies for managers to incorporate a responsible and ethic engineering perspective? As a result, the aims of this paper are:

- to describe the design of a first-year subject, "Personal and Professional Effectiveness" from an experiential and person-centred approach;
- to explore the suitability of using Covey's 7 habits professional approach for the development of soft skills.

The article is organized as follows. First, the design of the module is described, including the rationale of using the Covey's 7 habits, the learning outcomes, activities and assessment tasks. Second, we explain some activities carried out to develop the different habits in detail so that they can be replicated. Third, students' perceptions about the learning process are presented and discussed.

## 2. Module Design

The module was specifically designed to support the development of soft skills for engineering first year students. It is a six European Credit Transfer System (ECTS) first-year module called "Personal and Professional Effectiveness", offered to all the engineering degrees as one of the three transversal subjects mentioned above. It takes place over six hours/week per trimester. It aims to enhance students understanding of the impact of soft skills in their future career as sustainable engineers, and to emphasize the importance of lifelong learning, and an understanding of the humanities and ethics [8,30].

To work with the students using a person-centred approach [31] and an experiential learning methodology [32], the teaching staff decided to draw on "The 7 habits of highly effective people" classic book [33], widely used to train managers in general [34] and in sustainability in particular [35] for leaders in engineering and technology [36]. Covey focuses on arguing that changes and difficulties can only be afforded through a set of habits that professionals must have or practice to become effective. These changes are required to work the ethical responsibility from a sustainable perspective that every effective leader

should have [35]. The entire design follows the concept of the maturity continuum [33]. Just as maturity is the process that leads to growth and development, the continuum refers to the continuous incremental nature of progression and growth. As the child grows into adulthood, maturity levels change from a state of high dependence (you need others to get what you want) to one of independence (you get what you want through your own effort). However, according to Covey there is one more stage, which represents the highest level of maturity. He calls this stage interdependence, and it allows you to get what you want with the highest level of quality while cooperating with others.

So, the subject design intends to develop, in students, the following 5 competences related to the learning outcomes: ethics, social responsibility, teamwork, interpersonal communication and autonomous work. Based on the above, the relationship between Covey's 7 habits and soft skills was established as a starting point for the design of the module (Table 1).

**Table 1.** Learning outcomes of personal and professional effectiveness related to the 7 habits [28].

| Learning Outcomes/Skills | Habits |
| --- | --- |
| Upholding Ethical Standards and Demonstrating Social Responsibility/Ethics; Social Responsibility; Sustainability | Be proactive (1) Being with the End in Mind (2) |
| Learning and Self-Development/Autonomous Work | Put First Things First (3) Sharpen the Saw (7) |
| Working well with Other/Teamwork; Interpersonal Communication | Think Win/Win (4) Seek First to Understand, Then to Be Understood (5) Synergize (6) |

To acquire these learning outcomes, the subject was designed so that students worked on:

- understanding the importance of developing the Covey's habits to become highly effective engineers;
- increasing knowledge of one's own strengths and weaknesses in the context of teamwork through experiential activities; and
- using personal and technical resources to design, plan, and present a sustainable project to an audience in a group setting.

The design took into consideration three key aspects to achieve the learning objectives:

1. The inclusion of a variety of activities to respond to the multiple intelligences' paradigm and learning preferences of the students under a person-centred approach and sustainability criteria.
2. The activities were proposed to the students almost as a game, without making the objective explicit at the beginning of the work. At the end of each activity, the students themselves carried out a final self-reflection to make the learning meaningful.
3. The activities had similarities with those carried out in programs for professional training, with the necessary adaptations for first year students.

At the beginning of the course, to start with, students were informed about the module's learning outcomes, the teaching-learning methodology, and the assessment criteria.

Class sessions always started with five-ten minutes of body awareness dynamics, breathing techniques, and visualizations to create a calm and secure atmosphere [37,38]. This was followed by the introduction of theoretical concepts related to one of the 7 habits and the competence to be developed, relating it to their future job as engineers. Then the experiential activity that was going to be done to work the learning objectives which helped to understand and internalize it. Some of these activities were designed by the teachers incorporating the different expected learning outcomes [39–42]. Others were adapted from the ones used in managers' training at companies [43–47].

Debates on current topics and case studies were also used to enable the search for rigorous information, the exchange of views between teams, the development of discursive skills and the refutation of arguments [48]. Jenga or Lego Serious Play [49] work on the development of different skills through gaming [50]. Role-plays and drama techniques [51–53] enhance speaking skills and help to explore different language styles, registers, and personalities. The following section describes in detail some of the activities carried out to develop the different habits.

To achieve deeper learning, after each session, students write a reflective entry about the activities done into their personal journals. Promoting reflection during learning helps students to become more aware of processes at the metacognitive level; this increases the student's responsibility and motivation regarding their own learning process [17,54–56].

Connected with the activities, several procedures of assessment were used to establish if learning outcomes were achieved including debates and oral presentations, mind maps, reports, and personal diaries. Rubrics, checklists, and questionnaires were used for assessment. Table 2 shows the activities and assessment tasks with the percentage contribution to the summative grading for the final assessment.

**Table 2.** Learning activities and assessment tasks.

| Learning Activities | Assessment Tasks | |
| --- | --- | --- |
| Mind map<br>Individual reports<br>Blog participation<br>Exam on contents | Individual-based activities<br>(30%) | |
| Debates<br>Project report<br>Reflection about dynamics and games<br>Role-plays | Group-based activities<br>(30%) | Portfolio (30%) |
| Debates<br>Attendance to conferences | Interclass activities<br>(10%) | |

For each of Covey's habits, two or three different activities were carried out during the trimester. The examples shown in the following sections are intended to be a representation of the range of activities implemented during the course. For each one of the learning objectives of the activity, its description and the assessment is described.

### 2.1. Habit 1: Be Proactive. Activity: The Imagined House

For an engineer, beyond the goodness of being proactive per se, being able to make the right decisions, analyse every single option and evaluate all the consequences should be part of the expected sustainability competences to work on projects. Engineers must be aware that their projects, and therefore their decisions affect people's lives and managing risks must be always under control. That is the real importance of this habit in the training of future engineers. The purpose was that students took responsibility for their choices and behaviors. Proactive people focus their efforts on their circle of influence. This habit is built from a knowledge of one's own paradigms and principles (i.e., what is important and essential for us) prior to interacting with others.

Learning objectives were: to confront the students with new modes of communication; to make them reflect on aspects of their own personality; to practice listening, respect and consensus in a group containing individual differences; and to create confidence in the workspace and the group.

The activity consisted of several stages:

- Phase 1: Imaging the perfect house. Students with closed eyes were guided through a visualization to imagine a house they could design and furnish as they wished. After the visualization, the students could draw their house (either literally or symbolically)

with different colors and textures. They also wrote the description of the house with as many details as they could remember.

- Phase 2: Sharing and building a common house. In groups of 4–5 members, each student explained what their house was like. The rest were encouraged to listen without interrupting. From the individual contributions, the group created a common house, in which everyone could feel represented, safe and confident.
- Phase 3: Final debate and reflection. The group houses were shared with the whole class, fostering a common debate on meanings, values, desires, differences, and interests.

The assessment was done through individual students' reports. This report provided valuable information to the teacher about the student's ability to confront the challenges, level of self-knowledge, ability to listen to others, ability to defend important personal characteristics (projected in the image of the house), and acceptance of differences. Deeper and more elaborate meanings emerged during the post-session personal reflection.

*2.2. Habits 2 & 3: Being with the End in Mind & Put First Things First. Activity: How Would You Solve It?*

Habits 2 and 3 are probably the most valuable habits for a fresh engineering. First, because of the real need of being well organized and capable to prioritize tasks along time depending on effort, feasibility, and positive impact on the results. Secondly, because it implies the assumption of being fallible as a professional and, moreover, being able to react and steer the rudder to the best possible direction. Both skills are crucial in an engineering behavior [4,57].

So, the learning objectives were to identify an ethical and sustainable solution to a well-known problem (i.e., social, environmental, or economic); to establish solution-oriented objectives, tasks and resources; to organize the tasks in terms of importance, impact, and urgency and plan a schedule for their implementation; and, to present and defend the proposal. The dynamic is also thought to stimulate teamwork but, in addition, to enhance the listening skills and the ability to find a consensus.

The activity was organized through different phases:

- Phase 1: Identification of problems and possible solutions. Each student must individually think of three different current problems and possible solutions. The members of the team choose a problem from the different proposals and discuss possible solutions. They try to eliminate those that are redundant and merge those that are similar.
- Phase 2: Selection of objectives and actions. Once they agree, the group is asked to write down a general objective, 2–3 specific objectives, a list of actions to meet these objectives, and the resources needed.
- Phase 3: Organising priority. Once the actions are identified, a table is built with one row per action and 4 columns (Urgency, Feasibility, Efficacy, and Final Priority—Global average). Each team must fill out the table for every action assigning a value (1 to 3) in terms of Urgency, Feasibility, and Efficacy (see Figure 1).
- Phase 4: Writing the action plan. Each group should prepare a document containing the general and specific objectives, the actions according final priority, timeframe, and the resources needed to carry them out. Difficulties and obstacles must also be included.
- Phase 5: Defense. Each group present the plan to the colleagues. Other students are encouraged to ask questions or make suggestions that could improve it.
- Phase 6: Reflection. At the end of the activity, each group has time to improve their document based on suggestions or questions raised in class. They should include an overall reflection, emphasizing which aspects helped them to achieve good results and which did not.

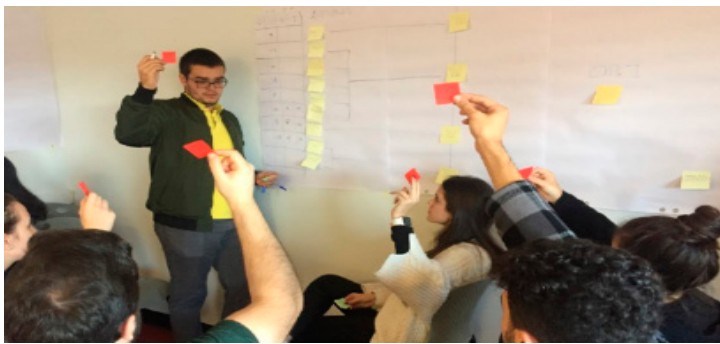

**Figure 1.** Students taking decisions in activity for habit 1.

A three-level rubric (see Table 3) was used to assess the learning outcomes.

**Table 3.** Rubric for the assessment of Activity "How would you solve it?".

| Indicator | 0 (Beginner) | 1 (Competent) | 2 (Very Competent) |
|---|---|---|---|
| Objectives | There's only one specific objective, it's too generic and it doesn't meet SMARTER | There are several specific objectives, but they don't meet the SMARTER criteria | All the specific objectives meet the SMARTER criteria |
| Obstacles/Barriers | There are any barrier or difficulty identified | There are some barriers named without discussion | Several barriers are identified and discussed. They are considered in the final plan |
| Actions | There are not enough actions identified by the group | Different actions are established, but some are repeated or have not been merged by affinity | There are enough clear actions as well as the dependency between them and the time in which they must be carried out |
| Urgency, Feasibility and Efficacy | There are no values assigned to these items | There are values assigned for these items, but the students haven't understood the difference between them | There are values for these items. The group could explain the reasons for those values. The group has understood the differences |
| Plan | At the end of the activity, there is not a clear plan | At the end of the activity, there is a plan with goals, actions and how to prioritize them. Some errors in the sequence of actions can be observed | At the end of the activity, there is a clear plan, actions are identified according to urgency, feasibility, and efficacy. Difficulties are also identified |

### 2.3. Habit 4: Win-Win. Activity: Cats & Mice

Engineering must be a discipline that seeks sustainable solutions to society's problems. Thus, the introduction of this habit aims to convey the following message: a solution is not good enough if someone must lose something because of it. The engineer must be aware of the need to find solutions that ensure that everyone wins and must be able to always negotiate to get there.

The learning objective was to learn the different negotiation styles (win-win, win-lose, lose-lose, and lose-win) becoming aware that to win it is not necessary for others to lose.

The activity is a game where the students are asked to earn as much money as possible just by choosing one of two cards ('Cat' or 'Mouse'). They play in teams (4–5 students each). All of them must raise the card at the same time and the money is earned or lost according certain rules. For instance: if each team raise a mouse, all of them earn money but if only one of them raises a cat, only the cat earns money and the rest loses it.

Ten rounds were played, but in certain moments between rounds, teacher could change the rules, to let them negotiate among teams, or change the payments table to use a multiplier to make the profit and lost bigger when the teacher so decided. Usually at the end of the tenth round many different scenarios have taken place: teams may have lost

lot of money, earned some money, failed to keep their agreements, or always chosen cats or mice.

During the reflection after the game, teachers should remind students that they were never asked to earn more money than the rest, only that they should earn as much money as possible. Students must realize that, if they had picked a mouse in each round, all the teams could have earned more money than the team that made the most money during the game. This is a good way for students to discover that their own profit is not always the best option and many times the best options implies that everyone may be able to earn something.

For the assessment, after the discussion in class, an individual report was asked of each student containing two parts. On one hand, the student should reflect on the team dynamics, observations of individual behaviors during the session, and thoughts about the win-win concept. On the other hand, the student must look for examples (real or fictional) that represent the different negotiation styles and explain them.

### 2.4. Habit 5 & 6. Seek First to Understand & Synergize. Activity: Jenga Game

Working with this habit offers engineering students the opportunity to make them aware of the importance of being able to put themselves in someone else's shoes. For an engineer it is essential to understand the difficulties of being a project leader and also those of being a member of a team, where the engineering project does not go ahead without the orchestrated work of everyone. Thus, when working on this habit, the aim is to work on the importance of understanding different sensitivities or strategies to successfully face a task.

The learning objectives of this activity were: to improve active listening, effective communication and empathy; to practice different roles and duties in a group; and to make students understand that synergy is about combining people's strengths through positive teamwork. Students should employ collaborative work to achieve a common goal in the most efficient way. This also means recognising their own strengths and learning about healthy group dynamics.

The activity is a Jenga game in teams with some adaptations. Each team is made up with 3–4 workers and two leaders. Three rounds are played, each one with two phases (e.g., training and competition). The training phase lasts 12–15 min and the competition lasts 2–3 min. Each group needs to build a tower while workers keep their eyes closed using a mask (Figure 2). The leaders guide them, but they cannot touch the blocks nor the workers. During training time, they can try anything they want, but before the start of the competition, each group must commit to reaching a certain height using almost all 45 blocks. Each team works to reach their own goal, not to compete with others.

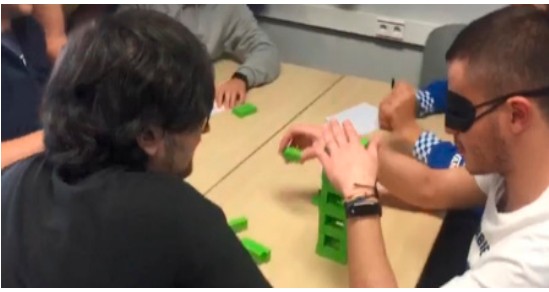

**Figure 2.** Students collaborating in activity for habit 5.

When the training phase is finished, the competition begins. During each round, the rules, roles, and goals should change slightly to explore how this affects the results. For instance, the objective can be bigger, the time can be reduced, or some changes in the members of the groups can be made. The results for every round must be recorded and compared. Finally, it is time to ask questions for reflection, e.g., what strategy did you follow to build the tower? Everyone can learn from the different strategies; which leaders

have used the mask during the training? Securely not, and that can help to explain the importance of empathy; which leaders have controlled the time? Surely a few leaders, and the responsibility is left to the teacher. How did you feel working with closed eyes? What have been your difficulties in guiding the work? What did you do differently in every round? How was the communication in the group?

To assess the learning outcomes of this activity two reports were asked for. One was a team report wherein students had to explain how they saw themselves as a team explaining the difficulties found in communicating, differences between leaders and workers, time management, and strategies. The second one was an individual report in which they were asked to reflect about their own strengths and skills, e.g., communicative styles, preferences between leader and worker, and stress management.

### 2.5. Habit 7: Sharpen the Saw

The learning objective of working on this habit was to be conscious about the importance of self-caring, that is, taking time to renew and refresh the following dimensions: physical, spiritual, mental, and social/emotional.

This habit was worked on in a transversal manner throughout the trimester with short readings and videos and small debates inserted into the activities of the rest of the habits. Additionally, the last activity of the trimester was a reading and synthesis of a scientific article on topics such as nutrition, rest, physical activity, and interpersonal relationships. This synthesis was reflected in the creation of a poster to be presented in front of the whole class.

The purpose of this activity for our engineering students was twofold. On the one hand, highlighting the importance of keeping our brain and body permanently prepared to act, learning from the articles they must read. But, in addition, teaching them how to stay updated by researching a certain topic that is crucial for a good engineer.

### 3. Method

In order to explore the suitability of using Covey's 7 habits in a professional approach to the development of soft skills, the perceptions of engineering students about the learning process used were gathered.

The target group consisted of 88 freshman (first year) engineering students. Being influenced by a person-centred approach to knowledge, a mixed method approach to data collection and analysis was used [58]. Mixing qualitative and quantitative data can increase the reliability and validity of the results and enhance their understanding and interpretation [59].

The findings relate to two distinct phases of data collection. The first phase took place during module delivery. Students' perceptions of the learning experience were gathered from the analysis of their personal portfolios. The second phase was carried out at the end of the trimester and was comprised of two elements:

- evaluating students' perceptions about the relevance and usefulness of the methodology, for which we designed a close-ended survey (Likert-type scales); and
- analyzing students' satisfaction, for which we used institutional satisfaction surveys. In this latter survey, only questions related to methodology, assessment, perception of learning and connection with the professional environment were considered. All these questions were close-ended; (Likert-type scales; one is the lowest grade and five the highest). This institutional questionnaire is standardized for all degrees at the authors' institution. Quantitative data were analysed through descriptive statistical analysis.

Students' perceptions about their learning process, methodology, and activities were qualitatively analysed from their diaries. Diaries are a good way to collect first-persons observations of experiences over a period. The aim of these diaries was to provide students with the opportunity to document their experiences, perceptions, and feelings about the subject and how it was developed. These reflections would provide students of significant learning experiences that have taken place during the session [60]. Students described their

feelings abundantly and openly in their diaries. First, systematic reading of all portfolios was completed, giving each student an individual code. This was followed by a deductive content analysis using the classic procedure described by Denzin & Lincoln [61]. The 7 habits were used as the initial categorisation matrix. For validity purposes, triangulation was achieved through the engagement of an additional researcher who was not involved in the teaching process for this qualitative analysis. NVivo software (version 12) was used for the coding process and analysis.

## 4. Results and Discussion

The results presented arise from the gathered data about students' perceptions of their learning and their satisfaction with the teaching experience triangulating them with the qualitative analysis from the students' diaries.

### 4.1. Quantitative Results of the Students' Perceptions

The survey about the relevance of the methodology used and the usefulness of the approach consisted of six questions.

The first statements included closed-ended questions (using a 4-point Likert scale from "Mostly agree" to "Mostly disagree"):

1. I have put into practice some of the concepts seen in the subject.
2. The subject has helped me to grow as a person.
3. In relation to the time spent in the classroom, to each content block (Ethics, Social Responsibility, Teamwork, Interpersonal Communication, and Autonomous Work), do you consider that the appropriate time was dedicated to each one? (Please, if you consider it, mark the blocks in which you would like a deeper knowledge).
4. The 7 habits seen in class are a good way to improve the competences of the subject. Then, two more questions with closed answers:
5. In relation to the methodology followed in this subject:
   a. I liked it more than the one used in others for its practical and experiential nature.
   b. I liked it, but less than the one used in other subjects.
   c. I think it didn't bring anything different compared to other subjects.
   d. I would've preferred another methodological approach.
6. Regarding the hours you dedicated to pass the subject:
   a. I had to work a lot more than in others.
   b. I had to work much less than in others.
   c. The workload was similar to others.
   d. I cannot assess this question.

The response rate was 61%. Analysing the results (Figure 3), we conclude that a high percentage of students (92.3%) think they have put into practice some of the concepts seen in the subject (51.9% agree and 40.4% slightly agree). Only 7.7% thought they haven't put into practice the concepts of the subject.

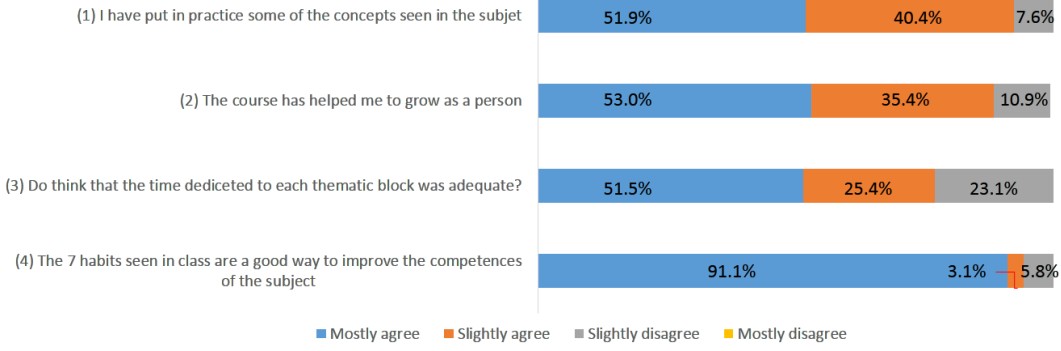

**Figure 3.** Students' perception results of the specific survey designed.

When students were asked if the subject helped them in their personal growth (question 2), 88.4% answered positively and 11.6% negatively. A significant number of students (94.2%) answered in question 4 that Covey's habits are a good way to achieve the module's outcomes. Most of them (76.9%) agreed on the amount of time spent on each part (question 3). A small percentage (23.1%) expressed the desire to have devoted more sessions to ethics and interpersonal communication.

Most students (86.3%) preferred the methodology used in this module (question 5) to more traditional ones. A small percentage (3.9%), however, would have liked to use a different methodological approach.

Finally, in relation to the time they dedicated to successfully complete the module (question 6), the responses were evenly spread: 19.2% considered they had to dedicate more time than in other subjects; 32.7% considered time spent to be similar to others; 32.7% considered that they spent less time than in others; and 15.4% did not respond to this item.

### 4.2. Quantitative Results of the Institutional Survey about the Teaching

Students' opinions about the aspects of their teaching–learning experience are collected by the university using a satisfaction feedback questionnaire with 20 questions. This Likert questionnaire (being 1 is the lowest grade and 5 the highest) is standardized for all university degrees. It aims to seek on several aspects of teaching, assessment, and support provided in each course.

For this study, we have only analysed questions related to the methodology, the assessment system and the perception about the learning. These are:

1. The adequacy of the teaching methodologies used to facilitate the learning of the subject is.
2. Assessment used in this subject is coherent with the activities and methodologies proposed.
3. The teacher relates the subject to the professional world.
4. With this teacher I have learned.
5. My overall judgment about the teacher is.

Results of the questionnaire show scores above 4 for all items: question 1 ($4.28 \pm 0.38$); question 2 ($4.19 \pm 0.4$); question 3 ($4.44 \pm 0.35$); question 4 ($4.01 \pm 0.43$); question 5 ($4.37 \pm 0.35$). The university mean for these items is 3.7 for each question.

### 4.3. Qualitative Results of Students' Perceptions

The qualitative data gathered were the personal diaries of the students during the course development. Students described their feelings abundantly and openly in their personal diaries. Texts were grouped and coded into categories. Each category was assigned to one of the module competences related to the 7 habits [28] according to a deductive approach. New categories emerged in the analysis. Then, diaries were re-read systematically to ensure that all aspects were covered. The thematic analysis was done with the help of the NVivo software (version 12). These categories (sub-categories in brackets) were:

- Upholding Ethical Standards and Demonstrating Social Responsibility (Habit 1, Be proactive; Habit 2, Begin with the end in mind);
- Learning and Self-development (Habits: 3, Put first things first; Self-knowledge; 7, Sharpen the Saw: continual improvement);
- Working well with Others (Habits: 4, Win-win; 5, Seek first to understand, then to be understood; 6, Synergize);
- Satisfaction with the teaching and with the subject.

#### 4.3.1. Upholding Ethical Standards and Demonstrating Social Responsibility

This theme encompasses students' recognition of the existence of ethical challenges in the profession and of the ned to maintain ethical standard despite external pressures to do otherwise. Students reported on how the activities done in the classroom influenced their learning. For instance, one student reflected about prejudice as "what really holds

you back when you meet someone" [Std4]. Another student elaborated on the value of having a clear ethical framework when she said, "Our emotions control our actions and our ethics justify them" [Std30].

The review of the diaries also indicated that students internalized the need to act responsibly to minimize the negative effects of their own choices on others, the community, and the environment:

- "if you don't agree with the ideas underpinning a project you must reject the project, because if you work for something that you do not like, you probably will not do it with much enthusiasm and the project will not go well" [Std16].
- "I learned the principle of commitment . . . sometimes within a company we must do things that really don't convince us, and yet we try to do our best" [Std3].
    - They learned to formulate their personal ethics, so, habits 1 and 2 were defined as subcategories within this general theme.

Habit 1: Be proactive.

Being proactive means to take responsibility for your choices and behaviors. Proactive people focus their efforts on their circle of influence. So being proactive they can translate their sustainability vision elsewhere [36]. Students appeared to have understood this habit and its importance. They identified proactive people as those who "work on the things they can do something about and there are many aspects of our life that we can influence" [Std4]. One student talked about the difficulty of recognizing the leverage points for action: "during the activity it seemed easy, but once I had to apply it in the project we had to develop in Physics and Maths I wasn't able to decide what I could control, even though I knew a good development of the project depended on my decisions" [Std13].

Habit 2: Begin with the end in mind

This habit relates to the imperative of beginning each day, task, or project with a clear vision of your desired direction and destination. Students perceived its importance when working on it: "you can get to study and dedicate yourself to what you want, if you map out a path to get there" [Std17]. As class sessions started with a relaxation exercise, when we worked this habit some students described that "this time helps me to disconnect and know how to begin with the end in mind" [Std4]. As Carlone analysed, becoming aware of one's own perceptions, emotions, attitudes and behaviors enables a commitment to change [34].

### 4.3.2. Learning and Self-Development

This theme refers to how people identify and address their own knowledge gaps and training needs. For instance, a student stated that "One of the things learned today was the amount of different views on the same topic and how someone can give up the beliefs held until that moment and change them with more interesting others" [Std2]. Students also reported being able to critically evaluate own strengths and weaknesses and thus proactively pursue their own development: "There were times when I didn't feel like it. . . because it was already taking me out of my comfort zone too much, for example with the debate, but I continued to overcome my fears and I improved my communication skills" [Std14].

Changes to how they were able to work better in a professional context were also highlighted: "This class, I believe, has helped us all to 'break the ice' and to open up so we can all work much better as a group" [Std13]. Another student explained how "body language expresses the attitude we adopt in different situations" [Std5]. They also described how they learned ways to provide feedback and coaching. In fact, they were "struck by how much we have improved, [ . . . ] as we can see, now we have much more confidence between us, and it's not difficult for us to give our opinion when working in groups" [Std24].

Habit 3: Put first things first

In relation to this habit, students learned to set priorities and manage their time so that the most important jobs are completed first, and not last: "We started with the internal

communication, about organizing our thoughts and our goals, then we were shown how to drive our thoughts towards fulfilling them" [Std36]. So, "the first thing to do in order to achieve a goal is to plan well" [Std16], but "I've seen, that to establish my priorities, I must overcome my fears and face them" [Std4].

Habit 7: Sharpen the Saw: continual improvement

It is interesting that, with regards to this habit, students not only described the need to constantly improve, but also the requirement to not comparing themselves to others. This was expressed as "we always must do things the best we can, to feel satisfied with our work and improve whenever possible. To get what we want we just must do what is necessary and believe in us" [Std27], but "we cannot get stuck or settle for something simply because everyone else does it or sees it that way, we must always seek change and improvement" [Std18]. They related this habit with effective teamwork: "An effective team is one that always tries to find new ideas and that has a spirit of self- improvement, is not satisfied with doing the minimum" [Std10]. A parallel can be found with the lifelong learning that Qadir & Al-Fuqaha emphasize as fundamental to an engineer's development [62].

### 4.3.3. Working well with Others

Teamwork is a fundamental aspect of engineering projects. Being able to listen to others' points of view makes it possible to develop creative ideas that benefit the whole team [63].

Through the activities done in the classroom, students identified positive interdependence and communication as an important factor for effective teamwork:

- "I must improve my social and communication skills, from the basic concepts of effectiveness and efficiency to the tools that help us stand out on a professional level" [Std3].
- "I've been able to realize that we must listen to others in order to learn" [Std27].

When changing the members in a team (in the Jenga game activity), they didn't feel comfortable: "maybe what has not worked well has been the fact of adapting to the way other people work when, within the group itself, we already knew how to work" [Std25].

Habit 4: Win-Win

Students found that win-win is a frame of mind where mutual benefit is constantly sought (in the Win-win activity). It is about working effectively with others to achieve optimal results: "it means getting what you want, achieving your dreams and being satisfied with the results" [Std23]. They realized that working win-win reduces the stress and identified the importance of a good planning: "because if everyone knows what he has to do and how to do it, everything is much easier than if everyone does things their own way without consensus within the team" [Std25]. To plan and identify the leader and workers roles is important as, otherwise, the chain may not work: "Being a leader, from my point of view, entails many more responsibilities, but also, you must transmit calmness and encouragement to the workers, which sometimes is not easy . . . Another detail to consider is communication between the leader and the worker. The leader has to put himself in the worker place and try to understand him, just as the worker must trust the leader, so that this chain work goes well" [Std45].

Habit 5: Seek first to understand, then to be understood

Related with communications skills, students perceived "that we must know how to listen to others so that we can all learn, and this shows how important communication is in a class as well as in everyday life" [Std25].

Habit 6: Synergize

Synergy means that the whole is greater than the sum of the parts. Its essence lies in valuing differences, in respecting them, in compensating for individual weaknesses, and in building on strengths (see the How to solve it? activity). Students described how "important the diversity of people in a team is, because in this way, many different ideas are provided" [Std10]. Moreover, how if there wasn't synergy, it could bring the projects to failure: "We had contrary thoughts and instead of trying to debate, we entered into some

kind of battle. I came to feel a little helpless with the situation and I hope that in future debates we'll improve" [Std14].

### 4.3.4. Satisfaction with the Teaching and the Subject

Once the marks were set, we asked the students to complete their diary with a final thought about the module (100% did it). This led to the emergence of a new category. The student reflections indicated that they were happy with the experience. Negative thoughts didn't arise in the portfolios. The only one was a student who wrote: "at the beginning it was hard to write something every day. It seemed tedious and a little waste of time. But finally, what it remains is that in the future it will help us to better understand some behaviors, both in daily life and in business life" [Std11].

Another student pointed out "to be sincere, the first day I didn't like it as the teacher asked us to go out of our comfort zone" [Std5].

Another one complained about the workload: "I think we've been asked to make too many activities outside the classroom and, sometimes, very dense, in addition to being in groups which sometimes meant more work because incompatibilities" [Std37].

Some students expressed a lack of understanding of the purpose of some of the activities done: "I did not understand very well the cat-mouse game. The Jenga game, the Lego, and others, however, I loved and learned a lot" [Std14]. Nevertheless, none of them indicated a dislike of the module or that it did not help them understand the value of the skills developed.

Some comments were even emotionally charged: "these months of class have been of immense help for my personal and professional development, as well as to improve my social and communication skills, starting from the basic concepts of effectiveness and efficiency to giving us tools that help us stand out at a professional level" [Std9].

In the main, students were very appreciative of the module: "It is not a subject that you usually have throughout the development of a person and, therefore, I think it is necessary to do something that is not usually taught, and you have done so. Thanks!" [Std12]; "This has been the only subject of the semester that I really wanted to attend, because I always had fun and I learned something... Every day of class I was waiting for what you would surprise us with in the next, because that is how this has turned out: surprising and unpredictable... I will never forget everything you have taught me, and I will never forget you. Thanks for everything" [Std17]. "Described in a few words: productive, fun, necessary, and effective. The subject personal and professional effectiveness has been highly influential in the organization of my work in the rest of the classes, as well as in family events and even sports" [Std21].

This is coherent with the results of Carlone [34], in which the application of the 7 habits results in a transformation of the self. When the 7 habits are combined with the soft skills-based learning objectives in a first-year engineering course, it becomes a powerful tool for personal growth. So, it seems that students felt that the approach of the course was appropriated.

## 5. Conclusions

In this paper the design of a competence-based first year subject from an experiential approach to incorporate a responsible and ethic perspective in their future as engineers has been described. The importance of using a person-centred approach to develop soft skills from the ideas in Covey's book has been outlined when looking for the impact this design has had in the students.

Survey results and diaries were analyzed to collect information about students' perception on the module design. This analysis shows that the approach used seems to have had a significant impact on students. Students' awareness of their own learning and the importance given to reflective thinking is supported by the content of their portfolios. Despite initial resistance, students' responses indicate that, over time, they appreciated the benefits of skills training for their future careers as engineers.

Some limitations in validity to qualitative analysis are apparent since students' diaries are self-reflections and discussions of the sessions produced in a situation where students knew that their teachers would read them. Thus, conscious or unconscious presumptions and a desire to please may have influenced their writing. However, the results, collected once the marks were given, corroborate the positive impact elicited from the analyses of the diaries. Qualitative results concerning their satisfaction with the teaching were favorable and supported by the quantitative findings. Indeed, a significant number of students emphasized that this approach is a good way to develop the desired competences. They also expressed their approval with the experiential approach as a key element in their learning.

In addition, students seem to have understood the importance of individual behavior in an engineering team: defining plan and goals, communicating, prioritizing and following up advances, leading people, and empathizing with clients or other colleagues. They feel that their mastery to deal with concepts like effectiveness, impact, influence, or leadership, among others, is key to success, and consider that having this subject in their curriculum important. They realize that ethical and social commitment must be part of the training to become an engineer. Moreover, the university-standardized questionnaire reflects a higher level of satisfaction with this module than the average. The worst evaluations are related to the assessment procedure and their perception of learning.

In sum, it seems that this person-centred approach allowed students to establish a deeper connection with themselves. They seem to have understood the importance of sustainability and how individual behavior impacts an engineering team. They expressed that being aware of their own strengths, difficulties, and opportunities allowed them to integrate new knowledge into their lives, not only professionally but also personally.

The design presented can be used as a guide for the design of new activities in the future.

**Author Contributions:** Conceptualization, R.-M.R.-J. and P.J.L.-B.; methodology, R.-M.R.-J., M.-J.T.-L. and P.J.L.-B.; software, M.-J.T.-L. and P.J.L.-B.; validation, R.-M.R.-J. and, M.-J.T.-L.; formal analysis, R.-M.R.-J., M.-J.T.-L. and P.J.L.-B.; investigation, R.-M.R.-J., M.-J.T.-L. and P.J.L.-B.; resources, R.-M.R.-J. and P.J.L.-B.; data curation, M.-J.T.-L. and P.J.L.-B.; writing—original draft preparation, R.-M.R.-J., M.-J.T.-L. and P.J.L.-B.; writing—review and editing, R.-M.R.-J., M.-J.T.-L. and P.J.L.-B.; visualization, R.-M.R.-J., M.-J.T.-L. and P.J.L.-B.; supervision, M.-J.T.-L.; project administration, R.-M.R.-J. and P.J.L.-B. All authors have read and agreed to the published version of the manuscript.

**Funding:** This research received no external funding.

**Institutional Review Board Statement:** Ethical review and approval were waived for this study, due to non-sensitive data were collected, data were properly anonymized and informed consent was obtained from all subjects involved in the study.

**Informed Consent Statement:** Informed consent was obtained from all subjects involved in the study.

**Data Availability Statement:** Not applicable.

**Conflicts of Interest:** The authors declare no conflict of interest.

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
