# Peer review of "Training Freshmen Engineers as Managers to Develop Soft Skills: A Person-Centred Approach"

_sustainability, doi:10.3390/su13094921_

Round 1
Reviewer 1 Report
The paper is composed by two main elements: a) the description of a subject design to train engineer students in soft skills throuh person-centered approach; and b) the analysis of students' perceptions about the effectiveness of training.
For the first element it is not clear the paper contribution, which seems related only to a detailed description of work organized by the university teaching staff. Section 2 proposes this description without citing references for the teaching staff who elaborated it, on the basis of “The 7 habits of highly effective people” classic book (line 89). If the design is already described elsewhere in terms of the 7 habits, which is the contribution of this section? It is not clear.
The second element seems to be more original and some qualitative and quantitative analises are proposed. Yet, it would be useful to further conceptualise empirical results, to give prominence to most relevant results. A synoptic table of results could be of major interest compared to the list of students' propositions.
In the whole paper, more emphasis is needed to the sustainability issue to enlight how the person-centered approach could help fostering sustainability behaviours.
In the introduction and in the conclusion sections must be clearly stated a comparison with similar analyses in the existing literature.
Author Response
The paper is composed by two main elements: a) the description of a subject design to train engineer students in soft skills throuh person-centered approach; and b) the analysis of students' perceptions about the effectiveness of training.
- Point 1: For the first element it is not clear the paper contribution, which seems related only to a detailed description of work organized by the university teaching staff. Section 2 proposes this description without citing references for the teaching staff who elaborated it, on the basis of “The 7 habits of highly effective people” classic book (line 89). If the design is already described elsewhere in terms of the 7 habits, which is the contribution of this section? It is not clear. The second element seems to be more original and some qualitative and quantitative analyses are proposed. Yet, it would be useful to further conceptualize empirical results, to give prominence to most relevant results. A synoptic table of results could be of major interest compared to the list of students' propositions.
Response 1: Thank you very much for your suggestion. We have tried to explain better our contribution, connecting the 7 habits with the competences that our students should acquire during the first year according our academic curricula. On the other hand, we give deep explanation about the structure of the lessons, type of activities and procedures of assessment.
Also some more references have been added to support the discussion of the results
- Point 2: In the whole paper, more emphasis is needed to the sustainability issue to enlight how the person-centered approach could help fostering sustainability behaviours.
Response 2: Thank you very much for your suggestion. We have tried to put more emphasis on the relationship between sustainable behaviours and person-centered approach in the introduction and more references have been added to support that.
- Point 3: In the introduction and in the conclusion sections must be clearly stated a comparison with similar analyses in the existing literature.
Response 3: We have tried to explain it better in the introduction. We have incorporated in section 4 references that support

Reviewer 2 Report
The paper Training Freshmen Engineers as Managers to Develop Soft Skills: A Person-Centred Approach describes a soft skills training course based on person-centered approach and on the classic book “The 7 habits of highly effective people”. It is an interesting and relevant research. 88 first year engineering students followed this course in this research. The aim of the paper is:
- to describe the design of the course and
- to explore the suitability of using Covey’s 7 habits professional approach for the development of soft skills.
The first aim of the paper is fully reached: the course design is excellent described and it has a strong and sound person-centered pedagogical design appropriate for the study, based on given literature and 7 habits professional approach.
Authors explore the suitability of approach based on the students’ perception using reflective diaries and by analyzing two surveys (one specific survey for the course and a general institution survey where only several specifically relevant question were used for the analysis). This is an appropriate method for this goal.
I have also some questions about this research and comments for improvements.
It is not clearly argued and no clear evidence given about what the exploration of the suitability of the chosen approach for the development of soft skills (the second aim) has shown us about the suitability of this approach for the students of engineering. The authors refer in the paper to “projects” but they don’t describe how the projects were connected to the engineering content knowledge or to the acting as being engineers. Thus it is not described weather or not this has made any changes in students’ attitudes within the engineering context and the context of studying engineering and preparing them for their work practice as engineers. Did they only reflected on games, i.e. designing a dream house, play mice and cats, etc.? If there is no connection with the engineering content or with acting as an engineer (context) than the results of this research cannot prove any suitability of the method for the engineering students and also not the needs to include such a course in the curriculum in their first study year. It is well known from the literature that in general this methods work for successful people.
Nevertheless, the line 108 suggests that there was a connection with engineering but except this sentence I couldn’t find any more explanation in the paper. Please explain more what have the students done here.
Line 108: “To use personal and technical resources to design, plan and present a sustainable project to an audience in a group setting.”
This needs to be improved in the paper or the authors need to clearly give the limitations of this research.
The students are positive about the course, about the methods and about the teachers in their reflective diaries. The quotes from the diaries are interesting to read and they give valuable insights in the reasoning of students. The quotes are also only anecdotical in their character and the authors have included only positive experiences of students. Were there also any negative experiences? Please provide more discussion about it and underpin it by quantitative data about the answers of 88 students. I understand from the paper that text was analyzed.
Author Response
- Point 1: The paper Training Freshmen Engineers as Managers to Develop Soft Skills: A Person-Centred Approach describes a soft skills training course based on person-centered approach and on the classic book “The 7 habits of highly effective people”. It is an interesting and relevant research. 88 first year engineering students followed this course in this research. The aim of the paper is:
- to describe the design of the course and
- to explore the suitability of using Covey’s 7 habits professional approach for the development of soft skills.
The first aim of the paper is fully reached: the course design is excellent described and it has a strong and sound person-centered pedagogical design appropriate for the study, based on given literature and 7 habits professional approach.
Authors explore the suitability of approach based on the students’ perception using reflective diaries and by analyzing two surveys (one specific survey for the course and a general institution survey where only several specifically relevant question were used for the analysis). This is an appropriate method for this goal.
I have also some questions about this research and comments for improvements.
It is not clearly argued and no clear evidence given about what the exploration of the suitability of the chosen approach for the development of soft skills (the second aim) has shown us about the suitability of this approach for the students of engineering. The authors refer in the paper to “projects” but they don’t describe how the projects were connected to the engineering content knowledge or to the acting as being engineers. Thus it is not described weather or not this has made any changes in students’ attitudes within the engineering context and the context of studying engineering and preparing them for their work practice as engineers. Did they only reflected on games, i.e. designing a dream house, play mice and cats, etc.? If there is no connection with the engineering content or with acting as an engineer (context) than the results of this research cannot prove any suitability of the method for the engineering students and also not the needs to include such a course in the curriculum in their first study year. It is well known from the literature that in general this methods work for successful people.
Response 1: Thank you very much for your positive comments. We have tried to explain better the connection between engineering context and each habit. The activities allow students to explore real-world situations that an engineer will find during his/her professional career. All of them pretend to make students reflect about the importance of sustainability and individual behavior in an engineering team. To support it some more references about it have been added.
Point 2: Nevertheless, the line 108 suggests that there was a connection with engineering but except this sentence I couldn’t find any more explanation in the paper. Please explain more what have the students done here.
Line 108: “To use personal and technical resources to design, plan and present a sustainable project to an audience in a group setting.”
This needs to be improved in the paper or the authors need to clearly give the limitations of this research.
Response 2: Thank you very much for your suggestion. We have introduced the connection with engineering for each habit and some references have been also included.
Point 3: The students are positive about the course, about the methods and about the teachers in their reflective diaries. The quotes from the diaries are interesting to read and they give valuable insights in the reasoning of students. The quotes are also only anecdotical in their character and the authors have included only positive experiences of students. Were there also any negative experiences? Please provide more discussion about it and underpin it by quantitative data about the answers of 88 students. I understand from the paper that text was analyzed.
Response 3: Thank you very much for your suggestion. Negative reflections were scarce, we have included those.

Round 2
Reviewer 1 Report
The paper has improved and deserves publication.
Reviewer 2 Report
The paper is improved on all weak places. Congratulations for this good work!